# Macronutrient Intake in Adults Diagnosed with Metabolic Syndrome: Using the Health Examinee (HEXA) Cohort

**DOI:** 10.3390/nu13124457

**Published:** 2021-12-14

**Authors:** Hyerim Park, Anthony Kityo, Yeonjin Kim, Sang-Ah Lee

**Affiliations:** 1Department of Preventive Medicine, Kangwon National University School of Medicine, Chuncheon 24341, Korea; phr0922@gmail.com (H.P.); akkityo@kangwon.ac.kr (A.K.); bbkeye1229@naver.com (Y.K.); 2Interdisciplinary Graduate Program in Medical Bigdata Convergence, Kangwon National University School of Medicine, Chuncheon 24341, Korea

**Keywords:** macronutrient intake, metabolic syndrome, the Korean health examinee (HEXA) study

## Abstract

Macronutrient intake is important in the prevention and management of metabolic syndrome (MetS). This study aimed to evaluate total energy and macronutrient intake of participants diagnosed with MetS at recruitment of the health examinees (HEXA) cohort, considering the plant and animal sources of each macronutrient. We included 130,423 participants aged 40–69 years for analysis. Odds ratios (OR) and 95% confidence intervals (CI) were estimated to evaluate the intake of macronutrients stratified by gender. Energy and macronutrient intake were estimated by linking food frequency questionnaire data to the Korean food composition database, and were calculated separately for plant and animal foods. Low energy (men: OR = 0.95, 95% CI: 0.92–0.98; women: OR = 0.97, 95% CI: 0.95–0.99), and fat intake (men: OR = 0.93, 95% CI: 0.90–0.96; women: OR = 0.80, 95% CI: 0.77–0.83) were observed. Only postmenopausal women had lower intake of total energy (OR = 0.95, 95% CI: 0.92–0.97), whereas low fat intake was observed in all women (OR = 0.80, 95% CI: 0.77–0.83). For carbohydrate intake, the OR were 1.14 (95% CI: 1.08–1.22) and 1.17 (95% CI: 1.08–1.27) among women in their 50s and 60s, respectively. Protein intake was low (OR = 0.90, 95% CI: 0.86–0.95; and OR = 0.88, 95% CI: 0.82–0.94) among women in their 50s and 60s, respectively. High intake of plant carbohydrates in women (OR = 1.16, 95% CI: 1.12–1.20), and plant protein in both genders (OR = 1.09, 95% CI: 1.05–1.13) were observed, but low intake of total energy, fat, and animal-source carbohydrates in both genders was also observed. Fat intake was low regardless of food source. In conclusion, high consumption of plant-source macronutrients, and low consumption of animal-source macronutrients was observed in Korean adults diagnosed with MetS. Attention should be directed to plant sources of carbohydrates and proteins when designing population interventions for metabolic syndrome reduction in Korea.

## 1. Introduction

Interconnected risk factors of metabolic syndrome (MetS), such as abdominal obesity, atherogenic dyslipidemia, raised blood pressure, and insulin resistance, increase the risk of coronary heart disease, other cardiovascular diseases (CVD), and type 2 diabetes mellitus (T2DM) [1,2,3]. CVD and T2DM are global health issues with a high prevalence, and an increasing burden of economics in public health [4,5]. MetS has a high prevalence in many countries [6,7], and it has gradually increased in Korea over the last 10 years [8,9]. Although the etiology of MetS is unclear, various factors, such as genetics, metabolism, the environment, and diet, may contribute to its development [10]. A recent review highlighted a possible interconnection between gut microbial dysfunction, non-alcoholic fatty liver disease (NAFLD), and colonic diverticulosis as possible underlying risk factors of MetS [11].

Diet is an important factor in the prevention and management of MetS [12]. Previous studies reported conflicting results on the association between macronutrient intake and the risk of MetS according to gender, geographical location, and food sources [13,14,15]. In 55–80-year-old men, and women with MetS, intake of total fat above the acceptable macronutrient distribution range (AMDR), and lower intake of energy and carbohydrates were reported [16]. Plant protein, especially soy protein and wheat gluten, when compared to animal proteins, were linked to a reduced risk of MetS components [14]. In a meta-analysis of observational studies, high carbohydrate intake was associated with increased odds of developing MetS [15]. The observational study using national survey data from the United States and Korea showed not only different percentages of energy intake from carbohydrates, protein, and fat between the two countries, but also an increased risk of MetS with high carbohydrate intake only in Korean men and women [17]. This finding was attributed to the high percentage of energy from carbohydrates in Korea compared to the US.

Differential association between protein intake was also observed in Korean adults, as was a positive association between animal protein and MetS components (abdominal obesity, low HDL-C, and hyperglycemia), and a protective effect of plant protein on blood pressure in men, whereas null findings were observed in women [18]. The percentage of energy from carbohydrates in men, and specifically refined grains, including white rice, in women were associated with MetS [19].

The patterns of macronutrient intake in the Korean population with MetS have been rarely reported. Therefore, this study aimed to identify the characteristics of total energy, carbohydrate, protein, and fat intake in participants diagnosed with MetS at recruitment of the health examinees (HEXA) cohort, considering the plant and animal sources of each macronutrient. We also examined whether macronutrient intake differs by diagnosis of each metabolic syndrome component.

## 2. Materials and Methods

### 2.1. Study Population

The Health Examinees (HEXA) Study is a community-based, large genomic cohort that was conducted in Korea from 2004 to 2013. Detailed information about the study design have been described in previous studies [20,21]. For the homogeneity and comparability of participants, a previous HEXA study was updated to the Health Examinees-Gem (HEXA-G) [22]. A total of 139,345 participants aged 40–69 years remained in the HEXA-G data after excluding withdrawers (*n* = 11) and participants from invalid institutions (*n* = 31,375). In addition, 8922 participants were excluded among the HEXA-G. These included participants with missing baseline information on hypertension, diabetes mellitus, or hyperlipidemia history (*n* = 600); missing data on blood pressure, fasting glucose, triglycerides, high density lipoprotein (HDL)-cholesterol, and waist circumference (WC) (*n* = 4540); and those with energy intake <800 or ≥4000 kcal/day in men, and <500 or ≥3500 kcal/day in women (*n* = 3782). Finally, 130,423 participants, of whom 43,850 were men and 86,573 were women, remained in the final sample (Figure 1). All participants gave informed written consent prior to participating in the study. The Institutional Review Board of the Seoul National University Hospital approved this study for statistical analysis (IRB No. E-1503-103-657).

### 2.2. Dietary Assessment

Dietary information was collected using a validated semi-quantitative food frequency questionnaire (FFQ) developed for the Korean Genome and Epidemiology study (KoGES) [23]. Trained interviewers assessed the intake of 106 food items consumed by participants over the past year. Food consumption frequencies were classified into nine levels (from “never” to “three times or more a day”), and portion size into three levels (one-half, one, one and a half servings). Energy, carbohydrate, protein, and fat intake were calculated using a food composition table developed by the Korean Rural Development Administration (RDA) [24]. Plant- and animal-based carbohydrate, protein, and fat intake were determined after classification according to food source. Classification of macronutrients according to plant or animal sources was based on the guidelines of the KoGES database [25]. According to the guideline, the weighted amount of constituent foods based on the representative recipe of each food item constituting the mixed dishes are given. These amounts were used to evaluate mixed dishes, and calculate plant or animal source macronutrients. The percentage of energy from each macronutrient was calculated using the standard conversion factors: carbohydrate 4 kcal/g; protein 4 kcal/g; fat 9 kcal/g.

### 2.3. Definition of Metabolic Syndrome

MetS was defined using the National Cholesterol Education Program Adult Treatment Panel III (NCEP-ATP III) [26], modified for the Asian guideline for WC [27]. Participants who met three or more of the following criteria were classified as having MetS: (1) WC ≥90 for men, and ≥80 cm for women; (2) triglycerides (TG) ≥150 mg/dL, or drug treatment for elevated triglycerides; (3) HDL-cholesterol ≤40 for men, and ≤50 mg/dL for women; (4) systolic blood pressure (SBP) ≥130 mmHg, diastolic blood pressure (DBP) ≥85 mmHg, or drug treatment for elevated blood pressure; and (5) fasting blood glucose ≥100 mg/dL, or drug treatment for elevated fasting blood glucose.

### 2.4. Covariates

Demographic and lifestyle factors, such as age, marital status, education, family income, occupation, body mass index (BMI), health-related behaviors, and macronutrient intake were included. Age was categorized into three groups: 40–49; 50–59; and 60–69 years old. Marital status, education, family income, and occupation were categorized into two groups each: marital status—yes (married, cohabit), no (single, separated, divorced, widowed, others); education—12 years, 12 years or more; family income—below $3000, $3000 and above per month; occupation—occupied (11 types of occupation, including senior officials, managers, and professionals, etc.), unoccupied (unoccupied, housewife, and others). BMI was calculated as weight in kilograms divided by the square of height in meters (kg/m^2^). Smoking status was defined based on cut-offs used in the Atherosclerosis Risk in Communities study [28], and the HEXA study [29]. Non-smokers were defined as those who had smoked less than 400 cigarettes during their lifetime. Former smokers were defined as those who had smoked more than 400 cigarettes during their lifetime, but did not smoke at the baseline. Participants who had smoked more than 400 cigarettes during their lifetime, and still smoked at the baseline were classified as current smokers. Non-current drinkers were defined as those who had never consumed any alcoholic drink during their lifetime, or those who did not consume alcohol at baseline. Current drinkers were defined as those who still consumed alcohol. Participants were asked to report the number of times in a week, and the duration they regularly engage in activities that result in body sweating. Regular exercise was classified as engaging in activities that caused body sweating (for at least 5 days a week, lasting at least 30 min per session). Menopausal status was classified as either pre-menopausal women who currently experience monthly menstrual cycle, or post-menopausal women who have gone a year without menstrual flow.

### 2.5. Statistical Analysis

The Mantel–Haenszel chi-squared test for categorical variables, and the general linear model (GLM) for continuous variables were used to analyze the demographic and lifestyle factors of participants according to quartiles of total energy intake. Categorical variables, such as marital status, education, family income, occupation, smoking, drinking, and regular exercise, were presented as percentage (%). Continuous variables, including age, BMI, energy, and macronutrient (carbohydrate, protein, fat) intake were presented as median and interquartile range (IQR). Multivariable logistic regression models were used to examine the characteristics of energy and macronutrient intake in participants with MetS diagnosis at recruitment after adjusting for age, BMI, education, family income, occupation, smoking, drinking, regular exercise, and energy intake. Results are presented as an odds ratio (OR) and 95% confidence interval (CI). *p*-value < 0.05 (2-tailed test) was considered statistically significant. All data were analyzed using SAS (version 9.4; SAS Institute Inc., Cary, NC, USA).

## 3. Results

Demographic and lifestyle factors of participants according to metS are presented in Table 1. Both men and women diagnosed with MetS were significantly older (men: OR = 1.03, 95% CI: 1.03–1.04; women: OR = 1.07, 95% CI: 1.06–1.07), and had higher BMI (men: OR = 1.51, 95% CI: 1.49–1.53; women: OR = 1.40, 95% CI: 1.39–1.42) compared to controls. In addition, both men and women with MetS were more likely to be current smokers (men: OR = 1.37, 95% CI: 1.30–1.45; women: OR = 1.34, 95% CI:1.18–1.53), had an occupation (men: OR = 1.20, 95% CI: 1.12–1.28; women: OR = 1.12, 95% CI: 1.08–1.17), were less educated (men: OR = 0.89, 95% CI: 0.85–0.94; women: 0.75, 95% CI: 0.71–0.79), and less likely to exercise regularly (men: OR = 0.85, 95% CI: 0.81–0.89; women: OR = 0.90, 95% CI: 0.86–0.94) compared to controls. Current drinking was more frequent among men with MetS (OR = 1.09, 95% CI: 1.03–1.16). Women with MetS were less likely to be current drinkers (OR = 0.80, 95% CI: 0.76–0.84), and were less likely to earn ≥30,000 US dollars (USD) a month (OR = 0.87, 95% CI: 0.83–0.91) compared to controls (Table 1). Socio-demographic and lifestyle factors of participants according to quartiles of total energy intake are presented in Appendix A.

Table 2 shows energy and macronutrient intake of participants according to MetS and MetS components. Total energy intake was lower in men and women with MetS (men: OR = 0.95; women: (OR = 0.97). Low fat intake was also observed in men (OR = 0.93) and women (OR = 0.80), but high carbohydrate intake, and low protein and fat intake were only observed in women (carbohydrate: OR = 1.14; protein: OR = 0.90; fat: OR = 0.80). 

The intake of macronutrients according to the baseline MetS diagnosis stratified by gender and age group are presented in Table 3, and corresponding sample sizes of each group are presented in Appendix A. Low energy (OR = 0.94) and fat intake (OR = 0.91) were only predominant among 50–59-year-old men. Among women in their 50s and 60s, a high carbohydrate intake (OR = 1.14 and 1.17, respectively), and a low protein intake (OR = 0.90 and 0.88, respectively) were observed. Moreover, low fat intake was observed in women in their 40s, 50s, and 60s (OR = 0.87, 0.81, and 0.78, respectively). The median intake of energy and macronutrient intake for each MetS component stratified by age group is presented in Appendix A, whereas the corresponding odds ratios are presented in Appendix A. 

The characteristics of each macronutrient intake from plant and animal food sources are presented in Figure 2 for men and Figure 3 for women. Women diagnosed with MetS consumed more carbohydrates from plants (OR = 1.16). However, low intake of animal carbohydrates was also observed in men (OR = 0.92) and women (OR = 0.91). In both genders, low intake of animal proteins was observed (OR = 0.97 and 0.91 for men and women, respectively), whereas high intake of plant protein was reported (men: OR = 1.05, women: OR = 1.09). On the other hand, participants diagnosed with MetS had low intake of plant fat (men: OR = 0.97; women: OR = 0.89) and animal fat (men: OR = 0.95; women: OR = 0.87).

The characteristics of macronutrient intake according to MetS components also varied according to the source of macronutrients. Men with high triglycerides had a high intake of plant carbohydrates and proteins (OR = 1.08 and 1.09, respectively), but low intake of animal carbohydrates, protein, and fat (OR = 0.92, 0.96, and 0.94, in order). Similarly, men with low HDL-cholesterol reported high intake of plant carbohydrates (OR = 1.15), and low intake of animal protein and fat (OR = 0.89 and 0.85, respectively). Men with high blood pressure had a high intake of plant protein (OR = 1.05), and a low intake of animal carbohydrates and fat (OR = 0.93 and 0.96, respectively). High intake of plant protein (OR = 1.05) and animal fat (OR = 1.05), but low intake of plant fat (OR = 0.94) was observed in men with hyperglycemia. On the other hand, women with high triglycerides had a high intake of plant carbohydrates and protein (OR = 1.13 and 1.06, respectively), but low intake of animal carbohydrates (OR = 0.95). Moreover, women with low HDL-cholesterol had high intake of plant carbohydrates and proteins (OR = 1.21 and 1.06, respectively), but low intake of animal protein (OR = 0.91). High blood pressure among women was characterized by high intake of plant carbohydrates and protein (OR = 1.08 and 1.10, respectively), but low intake of animal protein and fat (OR = 0.96 and 0.93, respectively). Women who had hyperglycemia reported low intake of animal carbohydrates (OR = 0.95). The median macronutrient intake from plant and animal sources for men and women are presented in Appendix A.

## 4. Discussion

This study was designed to evaluate the macronutrient intake of participants diagnosed with MetS at recruitment of the HEXA cohort. Total energy and fat intake were low in both men and women diagnosed with MetS. Carbohydrate intake was high in both genders, but protein intake was low only in women. Participants diagnosed with MetS had low intake of carbohydrates and protein from animal foods, but high intake of carbohydrates and protein from plant foods. On the other hand, fat intake was low in both men and women regardless of source. Moreover, macronutrient intake varied across diagnosis of MetS components, and this variation also depended on the source of macronutrient.

Consistent with our findings, MetS participants consumed lower total energy compared to non-MetS participants [30,31]. In a 2005–2007 survey, Koreans 40 years and above with MetS consumed less energy than those without MetS [32]. In contrast, high energy intake was observed among MetS participants [33,34], whereas no differences in energy intake were observed between MetS and non-MetS participants [35]. In our study, total energy intake might have been restricted by participants prior to MetS diagnosis because of changes in food intake even when at least one of the five MetS components were diagnosed. Notably, individuals with MetS had a higher BMI, and were older compared to non-MetS individuals in this study. Under-reporting of energy intake has been shown to be common in individuals who are overweight or obese, as well as the elderly [36,37,38]. Thus, participants with MetS could possibly have under-reported their energy intake. Nevertheless, considering that MetS individuals are usually obese and sedentary, caloric restriction and weight loss are major interventions for MetS management [39]. Our findings may suggest increasing awareness of the role of caloric restriction for weight loss in MetS management among Korean adults.

Reports have indicated excessive intake of carbohydrates at the expense of other macronutrients in elderly Koreans [40], and a positive association between high carbohydrate intake and risk of MetS in women [41,42,43,44,45]. A typical Korean diet is rich in refined grains, especially white rice and noodles. Studies from Asian countries have reported a positive association between frequent consumption of refined grains and MetS [46,47]. Excessive intake of carbohydrates in the form of refined grains increases the glycemic load, and leads to postprandial hyperglycemia, hyperinsulinemia, and insulin resistance, which plays a key role in MetS, dyslipidemia, obesity, and T2DM [48]. Women with MetS reported high carbohydrate intake, unlike men. The gender difference is consistent with the higher intake of carbohydrates in Korean women than men [43,49]. Thus, carbohydrate substitution, or improving the quality of carbohydrates will likely benefit Korean women. Similar to our findings regarding plant carbohydrates, women, but not men, with MetS consumed more energy from refined grains and white rice [50]. These results suggest that gender is an important factor in the choice of carbohydrate sources among Koreans suffering from MetS, as women consume more carbohydrates from plant sources. A Korean-style diet, mainly characterized by grains and vegetables, may be associated with elevated risk of MetS [50]. In contrast, low intake of animal carbohydrates among subjects with MetS might reflect overall low consumption of milk in the Korean diet [51], since animal carbohydrates mainly come from milk and dairy products. Several studies have reported low intake of milk and other dairy products among MetS cases [52,53,54,55].

Women with MetS consumed less absolute protein than those without MetS, and absolute protein intake decreased with age, suggesting that Koreans restrict their protein intake as they age. On average, Korean adults consume 75.0 g/d of proteins, much more than the RDA of 50 g/d [56], whereas inadequate protein intake is only evident among the elderly population [57]. The association of protein intake with MetS is inconclusive, with some studies reporting positive [58,59,60,61], and others reporting negative findings among [33]. A Korean study reported increased odds of abdominal obesity, hypertriglyceridemia, and high fasting glucose, especially in middle-aged women with low protein intake [62]. Since our study was a cross-sectional study, it is possible that Korean women under-reported their protein intake, or restricted protein intake after diagnosis of one of the MetS, such as high blood pressure, low HDL-cholesterol, and high triglycerides, as reported in the current study.

Regarding plant or animal protein intake and MetS, there are previous studies that either agree or disagree with our findings. Some cross-sectional and longitudinal studies have reported a positive association between animal protein and MetS risk factors [60], and a negative or null association between plant proteins and MetS components [33,63]. A cross-sectional study in Korea reported a positive association between animal protein and MetS and its components, and a negative association between plant protein with high blood pressure in men, but not women [18]. Another Korean study of elderly adults found an inverse association between both animal protein and plant protein with abdominal obesity [18]. Median intake of proteins from animals was 25.9 g/day in men, and 17.4 g/day in women, a range relatively close to that reported in our study. A study from China showed an inverse association between animal protein and blood pressure in women, but not men, and pointed at a low intake of animal proteins in women [64]. A recent prospective cohort analysis from Iran reported a negative association between plant protein, animal protein, red meat, poultry, and egg intake with metabolic syndrome [65]. Notably, mean total, and animal and protein intake were very low in the Iranian study, ranging from 3.56–9.12 g/day for animal, and 2.94–4.18 g/da for plants. Another study from the US showed a favorable association between animal proteins and blood pressure, but an unfavorable association with fasting glucose and waist circumference, yet, plant protein was associated with improved fasting glucose and WC [66]. Favorable effects of both animal and plant proteins on blood pressure were observed in the Framingham study [67], whereas a study from Australia reported that plant protein from grain, legumes, and nuts was inversely associated with MetS and WC, but animal protein, particularly fried chicken and red meat, was positively associated with MetS [63]. The dietary quality of different sources of proteins, and differences in protein sources and culinary practices might differ between different populations, resulting in inconsistent associations between animal- or plant-sourced proteins with MetS [18,65]. The positive association between plant protein and MetS may be due to the correlation of protein with carbohydrate intake [57]. In Korea, plant foods predominate carbohydrate supply, and women consume less animal proteins, and proteins from legumes and vegetables compared to men [18]. Korean studies have reported more intake of proteins from plants, predominantly grains with only 1/3 of proteins from animal sources [18]. The composition of protein diets in Korea differs from that in the West. In one study in the US, plant protein diets were characterized by increased intake of nuts, seeds, soy, and legumes, and less added sugars, and were positively associated with the Health Eating Index [68]. Animal-based proteins in the US contribute 10% of daily energy, whereas plant-based proteins contribute 4–6% of daily energy intake [68], and beef, chicken, pork, processed meats, and eggs are the most consumed sources of proteins in the US [69]. Meats consumed in Korea are mainly beef soup, chicken, and grilled pork, and they are commonly wrapped in vegetables. Although Korea is undergoing Westernization, Koreans seem to conserve their traditional diet dominated by rice and vegetables, and low in animal foods [70]

Fat consumption has gradually increased in Korea, but is still within the acceptable macronutrient distribution range (AMDR), and lower than that of the US [71,72]. Lower consumption of animal source foods, such as processed red meat, in middle-aged Koreans has been reported [73]. Moreover, total unprocessed meat, particularly poultry consumption was inversely related with CVD risk among middle-aged Koreans [73]. A semi-Western diet, characterized by relatively high intakes of animal foods, was associated with a low risk of low HDL cholesterol [44]. Energy from total fat was associated with lower prevalence of MetS in both men and women. Similarly, participants with MetS were more likely to consume a low percent of energy from fats [50]. Among individuals in the unbalanced Korean dietary pattern, individuals in the extreme consumption of this pattern had a lower percent of energy from fat, and increased odds of MetS [44]. The percentage of energy from fat was significantly lower in MetS participants compared to normal participants [44]. In contrast, dietary total fat positively influenced metabolic syndrome in Tehranian adults, but the mean percentage of calories from fat was 30% [74], which is higher than our study. Longitudinal studies did not find any association between total fat and MetS, but found inverse associations with vegetable fats, but not animal fats [75]. Inconsistencies in findings might be due to different amounts of fats consumed in other countries vs. Korea, or the study designs employed. Some longitudinal studies and systematic reviews have reported no association between MUFAs, PUFAs, and SFAs with MetS or CVD risk factors [75,76,77], suggesting that dietary sources of fatty acids, rather than individual fatty acids and total fat intake, should be considered in the interpretation of our findings. Culturally, meat dishes are usually consumed with vegetables, and it is notable that meat intake in Korea differs from the Western style [70], which could underlie differences in the characteristics of fat intake among MetS subjects in Western countries and Korea.

Results of our study should be cautiously interpreted because of several limitations. First, because we used a cross-sectional study design, we could not examine any causal link between macronutrient intake and the risk of MetS. Second, we could not exclude participants with a previous diagnosis of MetS components due to incomplete data on previous diagnostic information on all MetS components. In this cohort, 0.84% participants reported having a diagnosis of diabetes, hypertension, or dyslipidemia at baseline. However, we believe that excluding those results would not have changed our results. Notwithstanding, there is a possibility that MetS participants could have changed their lifestyle after diagnosis of any one of the MetS components: high blood pressure; diabetes mellitus; hypertriglyceride; low HDL-cholesterol; and abdominal obesity. Third, we used a food frequency questionnaire to assess dietary intake. Since the food frequency method relies on participant memory, dietary intake could have been over- or underestimated by participants. Fourth, current food and nutrient intake might differ from the intake assessed at recruitment of this study. Therefore, our findings might not reflect current intake and changes in dietary intake over time. Lastly, we could not evaluate the amount of alcohol consumed by participants because alcohol intake was not quantified in the HEXA study. However, several strengths could be noted. First, our results can be generalized to the entire Korean population aged 40–69 years, since we analyzed a large sample of participants. Second, we used an FFQ that was previously validated for this population [23]. Hence, it is possible that we reduced the measurement error in dietary measurement. Third, we stratified our analysis by dietary source of macronutrients to capture differential characteristics of macronutrients and MetS according to macronutrient source. Fourth, we adjusted for known confounders of dietary intake and MetS. Nevertheless, residual confounding cannot be completely excluded.

## 5. Conclusions

We found low intake of total energy and fat in both men and women diagnosed with MetS, whereas high intake of carbohydrates, and low intake of protein was only observed in women. Although people diagnosed with MetS have high carbohydrate, and low protein and fat intake, this characteristic varied according to the source of macronutrients. The intake of plant carbohydrates and protein was high, whereas the intake of animal carbohydrates and protein was low. On the other hand, fat consumption was low regardless of its source. Our results suggest that plant sources of carbohydrates and protein need to be particularly evaluated when designing population-based interventions for MetS reduction. In addition, caution should be exercised when advising the population to reduce total consumption of animal foods. Also, the findings of our study need further confirmation by longitudinal cohort studies with a long follow-up duration and repeated dietary intake assessment.

## Figures and Tables

**Figure 1 nutrients-13-04457-f001:**
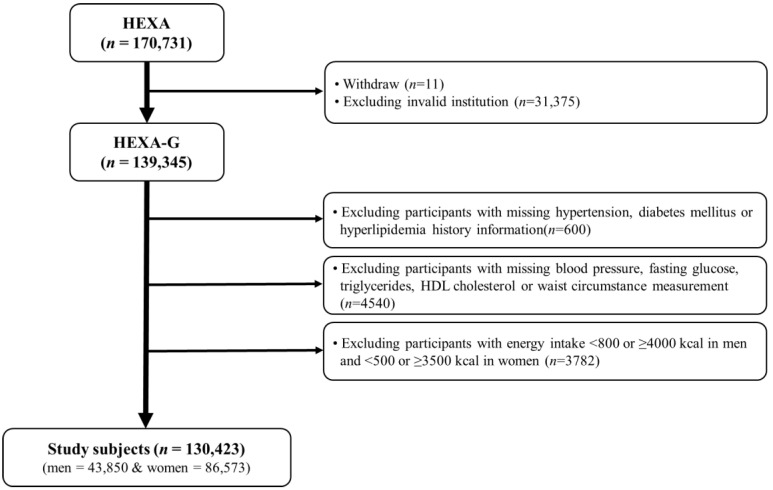
Flow diagram of analytical sample in this study using the health examinees (HEXA) cohort.

**Figure 2 nutrients-13-04457-f002:**
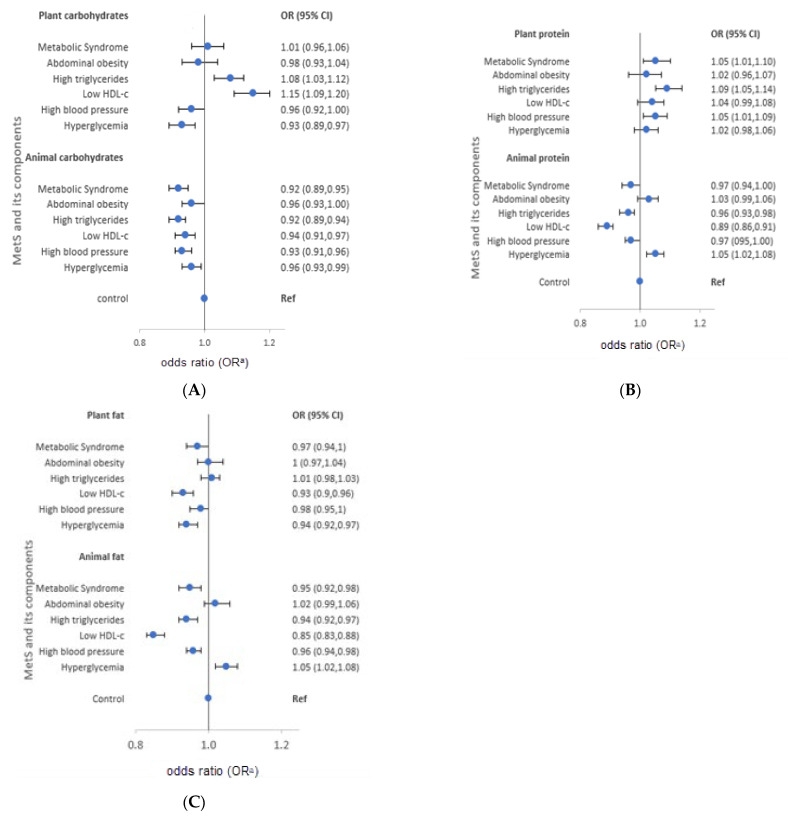
Odds ratios (OR) and 95% confidence intervals (CI) of metabolic syndrome and its components stratified by source of macronutrients in men, the HEXA-G study, 2004–2013. (**A**) plant and animal-source carbohydrates (**B**) plant and animal-source proteins (**C**) plant and animal source fats. HDL-c: High density lipoprotein cholesterol. ^a^ Adjusted for age, body mass index, marital status, education, family income, occupation, smoking, drinking, regular exercise, and energy intake.

**Figure 3 nutrients-13-04457-f003:**
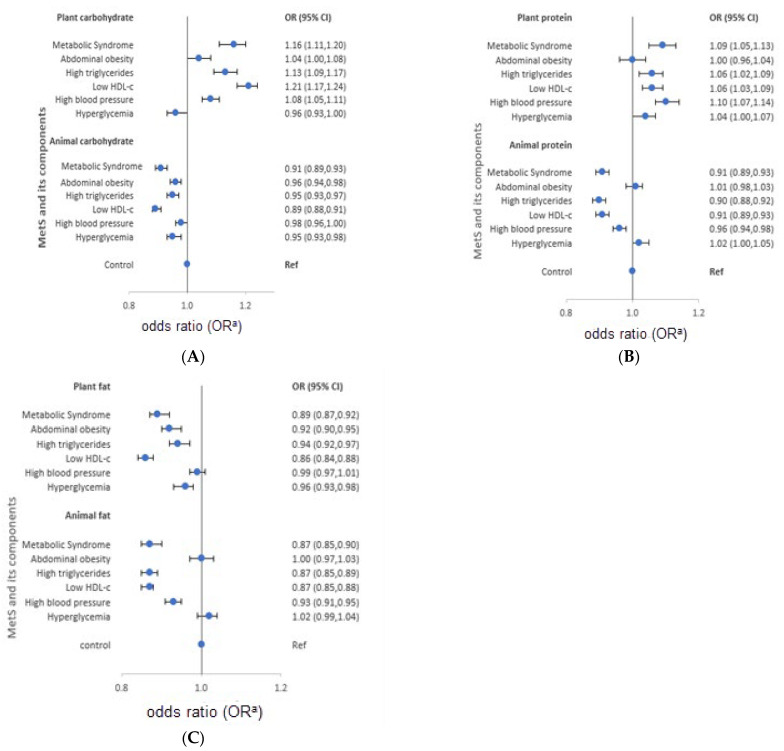
Odds ratios (OR) and 95% confidence intervals (CI) of metabolic syndrome and its components stratified by source of macronutrients in women, the HEXA-G study, 2004–2013. (**A**) plant and animal-source carbohydrates…(**B**) plant and animal-source proteins…(**C**) plant and animal source fats … HDL-c: High density lipoprotein cholesterol. ^a^ Adjusted for age, body mass index, marital status, education, family income, occupation, smoking, drinking, regular exercise, energy intake, and menopausal status.

**Table 1 nutrients-13-04457-t001:** Socio-demographic and lifestyle characteristics according to metabolic syndrome status, the HEXA-G study, 2004–2013.

	Men *(n* = 43,850)				Women (*n* = 86,573)	
	MetS (*n* = 12,640)	Control (*n* = 31,210)	OR (95% CI) ^b^	MetS (*n* = 21,028)	Control (*n* = 65,545)	OR (95% CI) ^b^
Age (years) ^a^	55 (48,61)	53 (46,60)	1.03 (1.03–1.04)	57 (51,62)	50 (45,56)	1.07 (1.06–1.07)
Body mass index (kg/m^2^) ^a^	26.0 (24.5,27.7)	23.7 (22.1,25.3)	1.51 (1.49–1.53)	25.4 (23.7,27.4)	22.7 (21.2,24.5)	1.40 (1.39–1.42)
Marital status						
No	5.7	5.8	Ref.	15.4	12.5	Ref.
Yes	94.3	94.2	0.94 (0.84–1.05)	84.6	87.5	1.04 (0.98–1.10)
Education (%)						
<12 years	58.8	55.0	Ref.	86.8	72.3	Ref.
≥12 years	41.2	45.0	0.89 (0.85–0.94)	13.2	27.7	0.75 (0.71–0.79)
Family income (%)						
<000 USD/month	51.4	49.9	Ref.	68.3	52.1	Ref.
≥3000 USD/month	48.6	50.1	0.95 (0.90–1.01)	31.7	47.9	0.87 (0.83–0.91)
Occupied (%)						
No	20.9	17.8	Ref.	67.0	58.2	Ref.
Yes	79.1	82.2	1.20 (1.12–1.28)	33.0	41.8	1.12 (1.08–1.17)
Current smoker (%)						
No	65.8	69.1	Ref.	97.6	97.7	Ref.
Yes	34.2	30.9	1.37 (1.30–1.45)	2.4	2.3	1.34 (1.18–1.53)
Current drinker (%)						
No	26.1	27.6	Ref.	76.5	67.1	Ref.
Yes	73.9	72.5	1.09 (1.03–1.16)	23.5	32.9	0.80 (0.76–0.84)
Regular exercise (%)						
No	44.7	42.1	Ref.	51.5	48.0	Ref.
Yes	55.3	57.9	0.85 (0.81–0.89)	48.5	52.0	0.90 (0.86–0.94)

^a^ Median (interquartile range, Q1, Q2). ^b^ Adjusted for age, body mass index, marital status, education, family income, occupation, smoking, drinking, regular exercise, and energy intake.

**Table 2 nutrients-13-04457-t002:** Odds ratios (OR) and 95% confidence intervals (CI) of metabolic syndrome and its components according to macronutrient intake, the HEXA-G study, 2004–2013.

	Energy (Kcal/day)		Carbohydrate (g/day)		Protein (g/day)		Fat (g/day)	
	Case/Control ^a^	OR (95% CI) ^b^	Case/Control ^a^	OR (95% CI) ^b^	Case/Control ^a^	OR (95% CI) ^b^	Case/Control ^a^	OR (95% CI) ^b^
Men								
MetS	1789 (1525,2125)/ 1781 (1516,2117)	0.95 (0.92–0.98)	318 (277,370)/ 317 (276,368)	0.97 (0.93–1.03)	59.5 (47,75)/ 58.4 (47,74)	0.98 (0.94–1.02)	27.4 (19,38)/ 27.3 (19,38)	0.93 (0.90–0.96)
Abdominal obesity	1821 (1546,2171)/ 1769 (1508,2100)	1.08 (1.05–1.12)	321 (279,377)/ 316 (275,366)	0.97 (0.91–1.03)	60.7 (48,77)/ 57.8 (46,73)	1.04 (0.99–1.10)	28.4 (20,40)/ 26.9 (19,37)	1.03 (0.99–1.07)
High triglycerides	1791 (1526,2129)/ 1778 (1514,2113)	0.97 (0.95–1.00)	318 (277,371)/ 317 (276,368)	1.05 (1.00–1.10)	59.3 (47,74)/ 58.3 (47,74)	0.98 (0.94–1.01)	27.7 (19,38)/ 27.1 (19,38)	0.94 (0.91–0.97)
Low HDL-c	1778 (1512,2109)/ 1785 (1520,2122)	0.94 (0.92–0.97)	319 (278,370)/ 317 (276,369)	1.14 (1.08–1.20)	57.9 (46,73)/ 59.0 (47,74)	0.87 (0.83–0.90)	26.3 (18,37)/ 27.6 (19,39)	0.81 (0.78–0.84)
High blood pressure	1776 (1514,2109)/ 1791 (1526,2132)	0.94 (0.92–0.97)	316 (276,366)/ 319 (277,372)	0.93 (0.89–0.97)	58.6 (47,74)/ 58.8 (47,74)	0.98 (0.95–1.02)	26.9 (19,38)/ 27.8 (20,39)	0.94 (0.92–0.97)
Hyperglycemia	1768 (1508,2100)/ 1791 (1524,2129)	0.95 (0.93–0.98)	315 (275,365)/ 319 (277,371)	0.91 (0.87–0.95)	58.8 (47,74)/ 58.7 (47,74)	1.07 (1.04–1.11)	26.9 (19,38)/ 27.5 (19,38)	1.02 (0.99–1.05)
Women ^c^								
MetS	1617 (1351,1924)/ 1652 (1359,1976)	0.97 (0.95–0.99)	300 (251,347)/ 300 (244,351)	1.14 (1.08–1.19)	51.8 (41,66)/ 54.0 (42,69)	0.90 (0.87–0.94)	20.9 (14,30)/ 24.0 (17,34)	0.80 (0.77–0.83)
Abdominal obesity	1643 (1367,1957)/ 1643 (1349,1967)	1.04 (1.02–1.06)	302 (251,350)/ 299 (242,350)	1.02 (0.98–1.07)	53.2 (42,68)/ 53.6 (42,68)	1.01 (0.98–1.05)	22.3 (15,32)/ 23.8 (16,34)	0.96 (0.93–0.99)
High triglycerides	1624 (1346,1940)/ 1648 (1360,1971)	0.99 (0.97–1.01)	300 (248,349)/ 300 (245,350)	1.11 (1.07–1.16)	52.0 (41,66)/ 53.9 (42,68)	0.90 (0.87–0.93)	21.3 (14,31)/ 23.8 (16,34)	0.84 (0.82–0.87)
Low HDL-c	1629 (1356,1943)/ 1652 (1358,1976)	0.97 (0.95–0.99)	300 (249,349)/ 300 (244,351)	1.17 (1.13–1.21)	52.4 (41,67)/ 54.1 (42,69)	0.91 (0.88–0.93)	21.6 (15,31)/ 24.1 (17,34)	0.79 (0.77–0.81)
High blood pressure	1619 (1346,1928)/ 1658 (1364,1983)	0.97 (0.96–0.99)	299 (247,347)/ 301 (245,352)	1.08 (1.04–1.12)	52.3 (41,66)/ 54.2 (42,69)	0.99 (0.97–1.02)	21.6 (15,31)/ 24.2 (17,34)	0.93 (0.90–0.95)
Hyperglycemia	1616 (1338,1927)/ 1649 (1362,1972)	0.96 (0.94–0.98)	297 (245,345)/ 301 (246,352)	0.93 (0.90–0.97)	52.3 (41,67)/ 53.7 (42,68)	1.04 (1.01–1.08)	21.7 (15,32)/ 23.5 (16,34)	0.99 (0.97–1.02)

HDL-c: High density lipoprotein. ^a^ Median (interquartile range, Q1, Q3). ^b^ Adjusted for age, body mass index, marital status, education, family income, occupation, smoking, drinking, regular exercise, and energy intake. ^c^ Additional adjustment for menopausal status in women.

**Table 3 nutrients-13-04457-t003:** Odds ratios (OR) and 95% confidence intervals (CI) of metabolic syndrome according to macronutrient intake by age group, the HEXA-G study, 2004–2013.

	Men			Women		
	MetS ^a^	Control ^a^	OR (95% CI) ^b^	MetS ^a^	Control ^a^	OR (95% CI) ^b^
Energy (kcal/day)						
40–49	1875 (1590,2241)	1846 (1568,2217)	0.96 (0.91–1.00)	1699 (1404,2044)	1703 (1393,2039)	0.97 (0.93–1.02)
50–59	1778 (1525,2111)	1774 (1511,2096)	0.94 (0.89–0.98)	1632 (1367,1937)	1635 (1342,1947)	0.97 (0.94–1.00)
60–69	1727 (1473,2032)	1711 (1471,2013)	0.97 (0.92–1.02)	1564 (1310,1842)	1571 (1308,1865)	0.97 (0.92–1.01)
Carbohydrate (g/day)						
40–49	327 (280,386)	324 (279,382)	0.99 (0.91–1.08)	307 (252,359)	305 (246,358)	1.05 (0.97–1.14)
50–59	316 (277,368)	317 (276,366)	0.95 (0.88–1.04)	302 (253,349)	299 (243,349)	1.14 (1.07–1.21)
60–69	312 (274,359)	311 (273,356)	0.99 (0.89–1.09)	295 (247,339)	293 (242,338)	1.17 (1.08–1.27)
Protein (g/day)						
40–49	63.1 (51,79)	61.4 (49,78)	1.00 (0.93–1.07)	55.7 (44,71)	56.1 (44,71)	0.99 (0.93–1.06)
50–59	59.3 (47,74)	58.3 (46,73)	0.96 (0.90–1.03)	52.4 (42,67)	53.3 (42,68)	0.90 (0.86–0.95)
60–69	56.4 (45,71)	55.1 (44,70)	0.97 (0.89–1.05)	49.3 (39,62)	50.0 (40,64)	0.88 (0.82–0.94)
Fat (g/day)						
40–49	31.5 (23,43)	30.7 (22,42)	0.94 (0.89–1.00)	25.3 (18,36)	26.5 (19,37)	0.87 (0.82–0.92)
50–59	26.8 (19,37)	26.8 (19,37)	0.91 (0.86–0.97)	21.2 (15,30)	22.8 (16,32)	0.81 (0.77–0.85)
60–69	24.4 (17,35)	23.9 (17,34)	0.94 (0.87–1.00)	18.4 (12,27)	19.6 (14,29)	0.78 (0.73–0.83)

^a^ Median (interquartile range, Q1, Q3). ^b^ Adjusted for age, body mass index, marital status, education, family income, occupation, smoking, drinking, regular exercise, energy intake, and menopausal status in women.

## Data Availability

All data and materials used in this study will be available upon reasonable request from the corresponding author.

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
