# Peer review of "Macronutrient Intake in Adults Diagnosed with Metabolic Syndrome: Using the Health Examinee (HEXA) Cohort"

_nutrients, 2021, doi:10.3390/nu13124457_

Round 1

Reviewer 1 Report

The aim of the present study was to explore the association of energy and macronutrient intake with Metabolic Syndrome (MetS) in a cohort of Korean adults aged 40-69. To investigate these associations, the authors used a large sample collected from the Health Examinee (HEXA) cohort which assessed energy and macronutrient intake using validated food frequency questionnaires. Briefly, the main findings were that low intake of total energy and fat in both men and women was associated with greater odds of MetS, whilst associations with high intake of carbohydrate and low intake of protein was evident in women only. Furthermore, these associations differed depending on the source of macronutrients whereby the intake of plant carbohydrates and protein was high and the intake of animal carbohydrates were low. In contrast, fat consumption was low regardless of its source. I would like to commend the authors on their impressive sample size obtained and their interesting findings highlighting geographical/cultural differences in their cohort of Korean adults compared to associations previously reported in Western cohorts. The novel aspects of their study are related to their Korean population, their analysis of the source of each macronutrient and whether this differs by each metabolic syndrome component. Overall, I believe minor revisions to the manuscript would be required for publication in Nutrients and I have listed some general and specific comments below:

General comments:

  1. There were a couple of potentially unexpected results compared to what is commonly observed in the literature and more words should be used in the discussion to fully explore these. The first is the finding that low intake of total energy was associated with greater odds of MetS in both men and women. It would typically be expected that total energy intake would be higher and the authors provide some discussion around participants potentially reducing their energy intake after MetS diagnosis. There is also the possibility that those with MetS potentially underreported energy intake to a greater extent than their non-MetS counterparts which would be consistent with previous literature suggesting greater underreporting in individuals with obesity and is particularly relevant given the greater BMI in those with MetS. Therefore, more discussion surrounding this issue should be provided in the second paragraph of the discussion.

  1. The second is the finding of higher intake of plant protein and lower intake of animal protein in individuals with MetS. It would typically be expected that animal protein may be associated with a greater risk of MetS and plant protein a lower or null risk, and the authors provide some references to these studies. In their discussion section, the authors partially allude to differences within the typical Korean diet as a potential explanation for this discrepancy which I am in agreement with. As these geographical/cultural differences are a key part of the novelty of the study, more words should be spent explicitly comparing and contrasting these dietary differences as a rationale for these findings. This comparison is done well in the following paragraph on the findings regarding fat intake.

  1. What was the authors’ reasoning for making age a categorical variable rather than keeping it as a continuous variable and examining age as an interaction term in the GLMs to see if it moderates the relationship between energy/macronutrient intake and MetS? I am not convinced the analysis of the different age categories adds to the meaningfulness of the data due to the relatively small age range of the cohort. I would suggest reporting the associations for men and women as a whole in the abstract rather than the data for the different age categories. I am also interested as to why the associations of low energy and fat intake were only present in men aged 50-59 years old. Could this be due a larger sample size in this group? It would be useful to present the sample sizes for the different age categories for the reader to examine.

Specific comments:

  1. Abstract – Line 13-15 – Provide more methodological information such as the male/female split and how energy and macronutrient intake were measured in the study.

  1. Abstract – Line 23-25 – The conclusion only currently restates the results section. An extra sentence on the relevance or wider implications of the findings would be useful.

  1. Introduction – Line 44 – Is 13 the correct reference for this statement? Reference 13 is a systematic review which includes participants with a wider age range than what is stated and also has no mention of lower energy intake in MetS.

  1. Methods – Line 120 – Is there a reference for these cut-off points for commonly defining non-smokers? If so, please provide.

  1. Methods – Line 127 – Was a certain number of exercise sessions per week used to define regular exercise? Please provide this information if so.

  1. Results – Line 177 – An odds ratio is missing for the data.

  1. Discussion – Line 264 – A reference is needed for this statement.

  1. Conclusion – Similar to my previous comment about the abstract conclusion, more words are needed around the relevance and wider implications of the findings. What do they mean for clinicians and what should they be advising for the prevention and management of MetS?

  1. Supplementary Table 1 – The title of the table does not match the table as there are no odds ratios or confidence intervals presented.

  1. Supplementary Table 2 – As mentioned previously, describing the sample size for each of the age categories would be useful to the reader.

Author Response

Please see response to comments

Reviewer 2 Report

The manuscript by Park et al analyzes the relation between metabolic syndrome and nutrient intake in a community-based, large genomic cohort that was conducted in Korea. There are some aspects that should be addressed.

  1. The authors should add the objective of the study in the abstract.
  2. The authors found that “Korean adults diagnosed of metabolic syndrome showed high consumption of plant-source and low consumption of animal-source macronutrients”. However, previous studies have shown the opposite. Furthermore, total energy intake was also lower in subjects with metabolic syndrome. The authors should discuss about it extensively because these findings were not expected.
  3. Why the authors excluded subjects with high energy intake (3500-4000 kcal)? How many were? I consider that these individuals should be included. This could be one of the reasons why patients with metabolic syndrome had lower caloric intake.
  4. In line with previous comments, the authors excluded participants who were undergoing treatment for any of metabolic syndrome components, since these participants could have altered their nutrient intake after diagnosis. How many patients were excluded because of this? Why the authors did not exclude all the subjects with previous diagnosis of metabolic syndrome who were not under treatment? It could also influence life style habits. It must be explained.
  5. All the data in the results section is included in the tables. The authors should add in this section just the most important findings.

Author Response

Please see response to comments

Round 2

Reviewer 2 Report

The authors have answered all the questions and have made some changes according to previous recommendations.

This manuscript is a resubmission of an earlier submission. The following is a list of the peer review reports and author responses from that submission.